# Water extract of ginseng alleviates parkinsonism in MPTP–induced Parkinson's disease mice

Ning Xu[1‡], Shuyang Xing[1‡], Jie Li[1], Bo Pang[2], Meichen Liu[1], Meiling Fan[3☉], Yu Zhao[1☉]*

1 Jilin Ginseng Academy, Changchun University of Chinese Medicine, Changchun, PR China, 2 College of Pharmacy, Jining Medical University, Rizhao, PR China, 3 The Affiliated Hospital of Changchun University of Chinese Medicine, Changchun, PR China

☉ These authors contributed equally to this work.
‡ NX and SX also contributed equally to this work.
* zhaoyu@ccucm.edu.cn

**Data Availability Statement:** All relevant data are publicly available from the figshare repository (https://doi.org/10.6084/m9.figshare.25397812).

**Funding:** This research was supported by grants from the Research and Development of Biological Macromolecular Components of Ginseng and

## Abstract

In this study, we investigated the neuroprotective effect of a water extract of ginseng (WEG) obtained via low–temperature extraction of the brain of mice with Parkinson's disease (PD) and the ameliorative effect on the damaged intestinal system for the treatment of dyskinesia in PD mice. MPTP (1–methyl–4–phenyl–1,2,3,6–tetrahydropyridine) was injected intraperitoneally into male C57BL/6 mice to establish a PD model, and WEG was given via oral gavage. The results indicated that WEG could protect the damaged neuronal cells of the mice brain, inhibit the aggregation of α-synuclein (α–Syn) in the brain, and increase the positive expression rate of tyrosine hydroxylase (TH). WEG significantly improved intestinal damage and regulated intestinal disorders (P<0.05). WEG intervention increased the levels of beneficial bacteria, such as *Lactobacillus*, and normalized the abundance and diversity of colonies in the intestine of mice. Our results suggested that WEG protected neurons in the brain of PD mice via inhibiting the aggregation of α–Syn in the brain and increasing the positive expression level of TH in the brain. WEG regulated the gut microbiota of mice, improved the behavioral disorders of PD mice, and offered some therapeutic effects on PD mice.

## Introduction

Parkinson's disease (PD) is a frequent chronic neurodegenerative disease of the elderly, manifested via movement disorders such as tremors, tightening of muscles, and bradykinesia [1]. The pathological features are mainly the accumulation of α–synuclein (α–Syn) in the patient's brain, the loss of dopaminergic neurons in the substantia nigra region of the brain, the reduction of tyrosine hydroxylase (TH) forcing the production of Lewy bodies, and the disturbance of the gut microbiota [2–9]. The latest research has found that PD may originate in the intestines. Since abnormal α-Syn first appears in intestinal neurons before the onset of PD, intestinal dysbacteriosis is most likely related to the onset of PD disease [10]. Intestinal

velvet antler in Jilin Innovation and Entrepreneurship Project, Project number: 2021Z003, Strategic Adjustment of the Economic Structure of Jilin Province to Guide the Capital Projects, Project number: 2014N155, and Science and Technology Development Project of Jilin Province, Project number: 20140101124JC. It is also supported by Health science and technology ability Improvement Project of Jilin Province, Project number: 2022JC035. The funder had no role in study design, data collection and analysis, decision to publish, or preparation of the manuscript. These grants provided material support for our study.

**Competing interests:** The authors have declared that no competing interests exist.

microorganisms act on the central nervous system through metabolites, neurotransmitters and cytokines secreted by immune cells, regulating brain-gut axis interactions [11], and the imbalance of the gut microbiota of Parkinson's patients may lead to a-Syn misfolding, from the intestinal nerves to the brain and brainstem, which in turn leads to brain dopaminergic neuronal damage [12]. Treatment for PD is often based on dopamine replacement therapy, which can only improve clinical motor symptoms but cannot cure the disease completely. Levodopa, for example, is converted to dopamine in the presence of Aromatic amino–acid decarboxylase [13]. As the disease progresses, however, the duration of levodopa's effectiveness is reduced and can also cause movement disorders. Data have suggested that some herbal extracts protect dopaminergic neurons and promote functional recovery from brain injury. Thus, herbs acting on multiple targets may have a potential role to play in the treatment of neurodegenerative diseases [14, 15].

Panax ginseng, a traditional herbal medicine, has been used as a traditional medicine in Asia for more than 2000 years. It is famous for its healthcare and therapeutic effects. Phytochemical studies have found that the main components of ginseng contain many active ingredients, such as polysaccharides, ginsenosides, and proteins [16]. Clinical studies have shown that ginseng has sedative, anti–aging, and neuroregulatory effects, as well as therapeutic benefits for neurodegenerative diseases [17, 18]. Modern research suggests that compounds in ginseng may be good drugs for treating neurodegenerative diseases, such as PD [19–21]. Jiang et al. studied the neuroprotective effect of boiling–water extract of ginseng on corticosterone–induced apoptosis in PC12 cells [22]. Van et al. found that alcoholic extracts of ginseng, such as G115, significantly reduced dopaminergic cell loss, microglia division, and the accumulation of $\alpha$–synuclein aggregates when administered orally [23]. Scholars have found that different ginseng chemical components have similar neuroprotective effects. Wang et al. conducted behavioral tests on mice and demonstrated that ginseng polysaccharides had antidepressant effects [24]. Ginseng polysaccharides also has been demonstrated to have antidepressant effects and may be closely related to neurotransmitters [25, 26]. Ginseng polysaccharides improved motor impairment in PD mice, as well as improved oxidative stress levels in mice, reduced the expression of apoptotic proteins in brain tissue of PD mice, and reduced parkinsonian symptoms [27]. Different types of ginsenosides exhibited neuroprotective effects in both in vivo and in vitro tests [28, 29]. Chen et al. found that ginsenoside $Rb_1$ restored neural loss by reducing oxidative stress and neuroinflammation, restoring cholinergic neuronal function, and rescuing cisplatin–induced memory impairment, suggesting that ginsenoside $Rb_1$ exerts neuroprotective effects [30]. At the same time, ginsenosides also have a potential therapeutic effect on intestinal microbiota dysbiosis. Ginsenoside $Rb_1$ significantly changed the composition of gut microbiota in hyperglycemic mice and increased the abundance of mucin-degrading bacteria *Akkermansia spp*. [31]. Ginsenoside $Rk_3$ intake caused significant changes in the intestinal flora of mice, enriching *Bacteroides*, *Alloprevotella* and *Blautia*, significantly reducing the *Firmicutes/Bacteroidetes* ratio, effectively improving intestinal dysbacteriosis and reducing the inflammatory response of mice. [32] Ginseng protein exerted neuroprotective effects on D–galactose/$AlCl_3$–induced hippocampus in rats [33]. Liu et al. demonstrated that ginseng protein prevented mitochondrial dysfunction and neurodegeneration in PD by ginseng protein giving *Drosophila melanogaster* PINK1 model of PD [34]. Numerous experiments have been conducted to demonstrate the neuroprotective effects of the three components—that is, ginseng polysaccharides, ginsenosides, and proteins. Whether the synergistic administration of these three components still had a neuroprotective effect and whether the WEG had an improvement on PD in vivo, however, have not been reported.

We know that ginseng extracts obtained using traditional extraction methods under high–temperature environments or with organic solvents. During high–temperature

extraction or in organic solvents, proteins in ginseng are decomposed and ginsenosides are partially converted under high–temperature conditions [35, 36]. In this study, we investigated the improvement of parkinsonism in mice using WEG obtained using low–temperature extraction. We used a 1–methyl–4–phenyl–1,2,3,6–tetrahydropyridine (MPTP) induced PD mice model to evaluate the therapeutic effect of WEG on PD mice according to behavioral indicators. For statistical analysis of MPTP–induced changes in $\alpha$–Syn and TH in mice brain, we observed the improvement of parkinsonism in mice brain by oral administration of WEG. By analyzing the sequence of gut microbiota of mice fecal samples, we addressed the changes in the gut microbiota in mice with PD. To evaluate the use of ginseng to treat PD, we studied the effect of WEG on the composition and diversity of microbial gut flora in PD mice.

## Materials and methods

### Animals and feeding

We used specific pathogen–free (SPF) male C57BL/6 mice (lot number) weighing $21\pm1$ g in this study. The animals were raised separately in a clean and dry laboratory with the following environmental conditions: temperature $25\pm2$°C, humidity $50\%\pm5\%$, alternating light and dark for 12 h/12 h, and free diet. All mice were purchased from YiSi Laboratory Animal Technology Co., Ltd. (Jilin Province, China). All of the animal experiments complied with the requirements of international experimental animal ethics, and were approved by the Animal Ethics Committee of Changchun University of Chinese Medicine.

### Establishing PD mice, grouping, and drug administration

The procedure to build the chronic PD mice model. was as follows: After 1 week of adaptive feeding, the mice were randomly divided into six groups (eight mice in each group), including control, model, positive control (P.C.), low, moderate, and high groups. We injected MPTP intraperitoneally every three days from days 0 to 14 (dose: 0.6 mg/mouse, blank group was injected intraperitoneally with equal volume of normal saline), and intraperitoneally every two days from days 14 to 28. Oral administration was given on days 28 to 56. According to the records in Pharmacopoeia of the People's Republic of China 2020, the daily oral dose of ginseng for human should be 3–9 g, and the corresponding administration groups of mice were categorized into low (0.013 g/kg body weight per day), moderate (0.026 g/kg body weight per day), and high groups (0.039 g/kg body weight per day).

### Preparation of WEG

The ginseng used in this study was purchased from Wanliang Ginseng Market (Jilin Province, China). Carbidopa and Levodopa Sustained Release Tablets (Carbidopa 50 mg and Levodopa 200 mg/tablet) were purchased from MSD (Hangzhou MSD Pharmaceutical Co., LTD., Hangzhou, China). To prepare the WEG, we crushed the ginseng herbs to form a powder, which was sieved and then placed in a conical flask with a stopper. Ginseng was extracted with neutral aqueous solution (pH $7.0\pm0.1$; ratio of ginseng powder to solvent 1:10) for three times for 12 h at 4°C. Using centrifugal supernatant fluid filtration, we obtained the samples after vacuum freeze drying. To maximize the retention of active components, such as protein and amino acid in ginseng, the temperature of the whole extraction process was kept below 30°C (the calculated extraction rate was $30\%\pm3\%$).

## Identification of main components in WEG

We determined the three main chemical components of the WEG. The total protein content of the samples was determined using a Bradford Protein Concentration Kit (Solarbio Science & Technology Co Ltd., Beijing, China), and the result was expressed as a percentage of total protein in the ginseng powder. The total polysaccharide content was determined using the phenol–sulfuric acid solution method, and the result was expressed as the percentage of total polysaccharide in ginseng powder. The ginsenoside content in the samples was determined using high–performance liquid chromatography (HPLC).

## HPLC analysis conditions

HPLC analysis was performed using Agilent LC–1220 HPLC (Agilent Technologies Co Ltd., Santa Clara, CA, USA), equipped with a binary solvent delivery pump, an automatic sampler, and an ultraviolet (UV) detector (Agilent Technologies Co Ltd.). The separation was performed on an Agilent SB–AQ C18 column (250 mm×4.6 mm, 5 μm). The flow rate was 1.0 mL/min. The detection wavelength was set to 203 nm. The column temperature was set at 35°C, and the injection volume was 10 μL. The mobile phase consistsconsisted of acetonitrile (A) and 0.1% phosphoric acid aqueous solution (B). The gradient program was as follows: 0–40 min, 19% A. 40–42 min, 19–22% A. 42–46 min, 26–32% A. 46–71 min, 32–38% A. 71–78 min, 38–49% A. 78–83 min, 49–51% A. All solutions were filtered using a 0.22–μm membrane filter before being added to the machine.

## Behavioral tests

**Pole test.** The pole climbing test is a classical method used to evaluate the motor coordination ability of mice. The climbing rod is an antiskid rod with a length of 50 cm and a diameter of 1 cm that is erected on a plane. The mice climbed from the top to the bottom of the rod, and the time spent climbing was recorded to compare their motor ability.

**Rotor test.** A rotor test was used to investigate the reaction and motor ability of mice. A rotating shaft was rotated at 8 g and mice were placed on the opposite side of the rotating rod and moved upward. Mice who stayed on the rotating rod, maintained their balance, and did not fall to the ground, maintained limb muscle coordination. Mice with motor problems or poor coordination fell more quickly. We recorded the amount of time the mice lasted on the rotating rod.

**Traction test.** A traction test is a method used to evaluate the motor ability of the mice. We took an 80 cm length of string with a diameter of 0.5 cm and fixed the ends of the string at the same height. The mice were placed upside down on the rope, and we recorded the hanging time.

**Morris water maze test.** The water maze test is a sensitive test used to evaluate the learning and memory ability of experimental animals to sense their position in space. The Morris water maze test used in this study was conducted in a circular pool (120 cm in diameter, 50 cm high wall). The pool was filled with water and the temperature was maintained at 24±1°C. We divided the pool equally into four areas. A circular fixed platform (10 cm in diameter) was submerged 1 cm below the surface of the water. On days 1 to 4 of training, the platform was moved to the same position (15 cm from the pool wall) and the mice were gently lowered from the edge of the pool wall at the four area edges (facing the pool wall). The starting position was in the water in the area opposite the platform. The experiment was automatically terminated when the mice climbed onto the platform for the first time and stood still for more than 3 s. If the mouse did not reach the platform within 120 s, the mouse was gently guided to the platform position. The fixed–position search test was conducted on day 5. We added titanium

dioxide to the pool to make the water appear white and opaque. We placed the platform in a fixed position 15 cm from the edge of the pool (submerged 1 cm below the surface of the water so that the mice could not see the platform due to the opacity of the water). We recorded the time when the mice found the platform and the actual swimming distance, as well as the swimming time of the mice in the inner and outer circles. The cruise experiment was conducted on day 6. We recorded the swimming time, stationary time, and swimming distance of the mice in 120 s. All data were obtained using a video tracking system (Taimeng Technology Co Ltd., Chengdu, China).

## Tissue collection

After completing the behavior test, the mice were euthanized after being anesthetized with carbon dioxide. The head and body were separated, and the rectum with fecal contents was immediately dissected and taken from all mice and placed in cryopreservation tubes. The samples were quickly placed in liquid nitrogen and stored at ultralow temperatures for the detection of gut microbiota. The mouse brain was quickly removed on a cold wooden board and immediately immobilized in 4% paraformaldehyde for 48 h for immunohistochemistry (IHC) and hematoxylin and eosin (H&E). Small intestine was taken from mice at the same location and immediately fixed in 4% paraformaldehyde for 48 h for H&E.

## Western blot analysis

The mice brains were lysed using RIPA lysis buffer (Solarbio Science & Technology Co. Ltd., Beijing, China) and the supernatant was collected using $10^4$g centrifugation at 4°C for 10 min. We determined protein concentration using the Bradford Protein Concentration Kit by adding the appropriate amount of loading buffer and then boiling the sample to denature it. Protein samples (30 µg per lane) were separated using gel electrophoresis and transferred to polyvinylidene fluoride (PVDF) films (Millipore, Bedford, MA, USA). After the transfer, the transferred PVDF films were placed in an antibody incubation kit and closed at 25°C for 2 h on a shaker with an appropriate amount of 5% skimmed milk powder, after closure the films were washed three times with Tris–buffered saline with polysorbate 20 (TBST) for 5 min each. PBST was discarded and the films were incubated overnight on a shaker at 4°C with primary antibodies against α–Syn (1:1000), TH (1:6000), glyceraldehyde 3–phosphate dehydrogenase (GAPDH) (1:1000), and β–tubulin (1:2000) (all antibodies were purchased from Abcam Biotechnology Co Ltd., Cambridge, UK). This process was followed by incubation with horseradish peroxidase–conjugated antibody for 1 h. We used β–tubulin and GAPDH as a loading control.

## Intestinal microbiome analysis

We conducted gut microbe 16S rRNA sequencing and data analysis. According to the manufacturer's protocol, the total DNA used was fecal DNA Isolation Kit (MP Biomedicals, Santa Ana, CA, USA). The amplified polymerase chain reaction (PCR) products were sequenced on Illumina MiSeq, and the high–quality sequences with 97% homology similarity were classified by operational taxon (OTU). The sequencing primer was F:ACTCCTACGGGAGGCAGCA R: GGACTACHVGGGTWTCTAAT and the sequencing region was 16S–V3V4. We used QIIME2 (Quantitative Insights Into Microbial Ecology, V2.0, https://docs.QIIME2.org/) and R package 3.5.1 (https://www.r-project.org/) for the 16S rRNA gene sequencing analysis. We analyzed alpha diversity by Chao1 and Shannon indices. Beta diversity was studied using principal coordinate analysis (PCoA) and a weighted UniFrac phylogenetic distance matrix. We determined

bacterial community structure using relative abundance at the phylum level and compared the differences between groups at the genus level using the heat map method.

## Statistical analysis

We plotted all data by GraphPad Prism 7.0 software and analyzed the data using SPSS 22.0 statistical software. Data were expressed as mean±SD. We used one–way analysis of variance (ANOVA) to compare the mean across multiple groups, and Tukey's test for the postcomparison of multiple groups. P<0.05 was regarded as the threshold of significance.

## Results

### Main compound mass in WEG

The calculated extraction rate of WEG was 34.23±3.50%. The protein content of WEG was 21.51%, accounting for 7.36% of ginseng:

$$Y = 0.6136X + 0.5851, \qquad (1)$$

where range = 0–1.5 mg·mL$^{-1}$, and R = 0.9967.

Ginseng polysaccharides in WEG was 68.40%, accounting for 23.41% of ginseng:

$$Y = 0.6327X + 0.1083, \qquad (2)$$

where range = 0–0.6 mg·mL$^{-1}$, and R = 0.9998.

WEG was quantified for the analysis of ginsenosides by HPLC. HPLC profiles are shown in Fig 1. Each ginsenoside contained the following: Ro 0.13%, $Rg_1$ 0.66%, Re 2.39%, Rf 0.27%, $Rg_2$ 0.09%, $Rh_1$ 0.03%, $Rb_1$ 4.93%, Rc 1.05%, $Rb_2$ 0.21%, $Rb_3$ 0.25%, Rd 1.15%, and $Rg_3$ 0.11%. With a total content of 11.27% for all ginsenosides, accounting for 3.86% of the ginseng. All standard substances were purchased from Yuanye Bio–Technology for (Yuanye Bio–Technology Co Ltd., Shanghai, China).

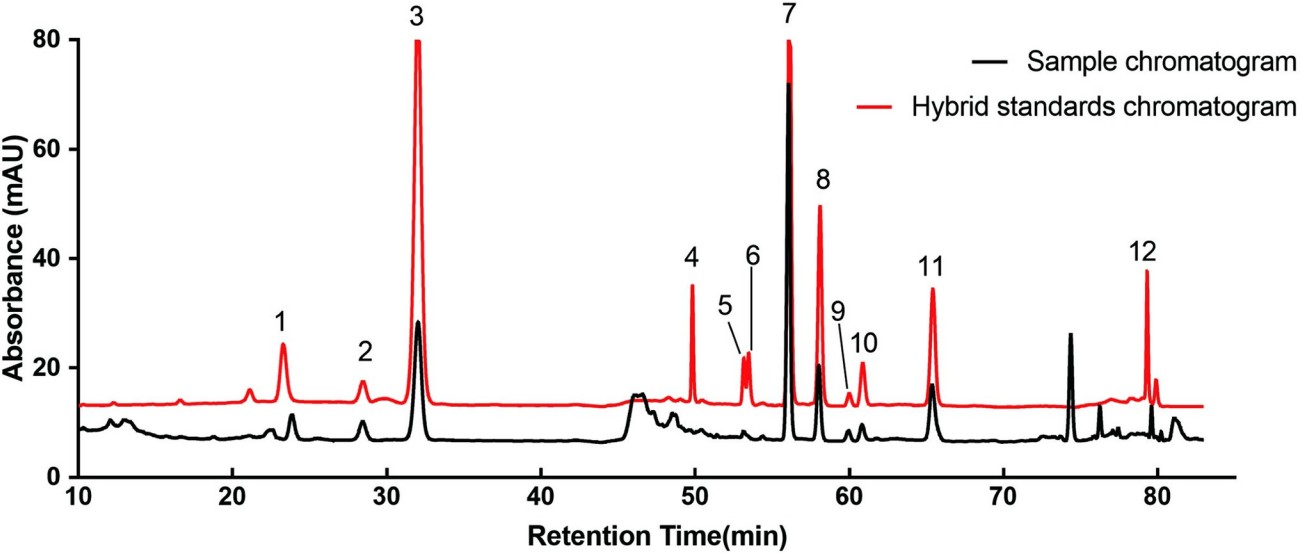

**Fig 1. HPLC chromatogram of ginsenosides (1: Ro. 2: Rg1. 3: Re. 4: Rf. 5: Rg2. 6: Rh1. 7: Rb1. 8: Rc. 9: Rb2. 10: Rb3. 11: Rd. 12: Rg3).**

**Table 1. Regression equation for 12 compounds.**

| Anlytes | Regression Equation | Correlation Coeffcient(R) | Range(μg) |
|---|---|---|---|
| Ro | Y = 593434X-3.6976 | 0.9995 | 0.16–0.97 |
| Rg$_1$ | Y = 133991X+21.999 | 0.9989 | 0.23–1.35 |
| Re | Y = 235757X+162.72 | 0.9988 | 3.33–19.98 |
| Rf | Y = 339669X-48.157 | 0.9975 | 0.25–1.52 |
| Rg$_2$ | Y = 377672X-2.3765 | 0.9997 | 0.09–0.52 |
| Rh$_1$ | Y = 521556X-2.3441 | 0.9994 | 0.08–0.47 |
| Rb$_1$ | Y = 194604X+28.418 | 0.9999 | 1.98–11.88 |
| Rc | Y = 219980X+5.8826 | 0.9997 | 0.86–5.13 |
| Rb$_2$ | Y = 171807X-0.375 | 0.9999 | 0.07–0.44 |
| Rb$_3$ | Y = 214806X+6.6436 | 0.9998 | 0.21–1.28 |
| Rd | Y = 249009X-7.1487 | 0.9998 | 0.60–3.60 |
| Rg$_3$ | Y = 529942X-7.9875 | 1.0000 | 0.16–0.96 |

## Validation of analytical methods

We evaluated the proposed method according to the working curves and analytical performance, including linearity, precision, accuracy, and stability:

- Linearity: The working curves were plotted from the peak area values against the concentrations of the 12 ginsenosides in the spiked samples (Table 1).

- Precision: The RSDs of the peak areas of the 12 analytes showed that Ro 0.62%, Rg$_1$ 0.94%, Re 0.21%, Rf 0.89%, Rg$_2$ 1.28%, Rh$_1$ 1.54%, Rb$_1$ 0.78%, Rc 2.33%, Rb$_2$ 1.05%, Rb$_3$ 0.60%, Rd 1.33%, and Rg$_3$ 1.38% were all less than 3%.

- Accuracy: The RSDs of the peak areas of the 12 analytes showed that Ro 0.91%, Rg$_1$ 0.68%, Re 1.21%, Rf 0.47%, Rg$_2$ 1.36%, Rh$_1$ 1.76%, Rb$_1$ 1.73%, Rc 2.06%, Rb$_2$ 1.64%, Rb$_3$ 1.38%, Rd 2.13%, and Rg$_3$ 1.56% were all less than 3%.

- Stability (12 h): The RSDs of the peak areas of the 12 analytes showed that Ro 1.47%, Rg$_1$ 1.37%, Re 1.74%, Rf 2.64%, Rg$_2$ 2.33%, Rh$_1$ 1.84%, Rb$_1$ 0.67%, Rc 1.74%, Rb$_2$ 1.83%, Rb$_3$ 1.76%, Rd 1.50%, and Rg$_3$ 2.23% were all less than 3%.

## WEG improved motor behavior in PD model mice

The mice behavioral experiments were conducted in two parts, in which the pole test, the traction test, and the rotary test were judgments of the strength of the mice's limbs and were used as the basis for whether the mice had motor impairment. For the Morris water maze test, mice were examined for their spatial learning and memory abilities. We determined whether there was an improvement in Parkinson's behavior based on spatial memory ability.

## Comparison of action–behavioral disorders of the limbs in mice

The pole test can be used to evaluate the muscular endurance of mice. The climbing time reflected the fatigue level of mice. A short climbing time decreased the muscular endurance (Fig 2A). The model group showed a reduced pole climbing time compared with the blank group (P < 0.01). The behavioral performance of mice treated with Carbidopa and Levodopa CR Tablets (CLCRT) or WEG was attenuated compared with mice in the modeling group.

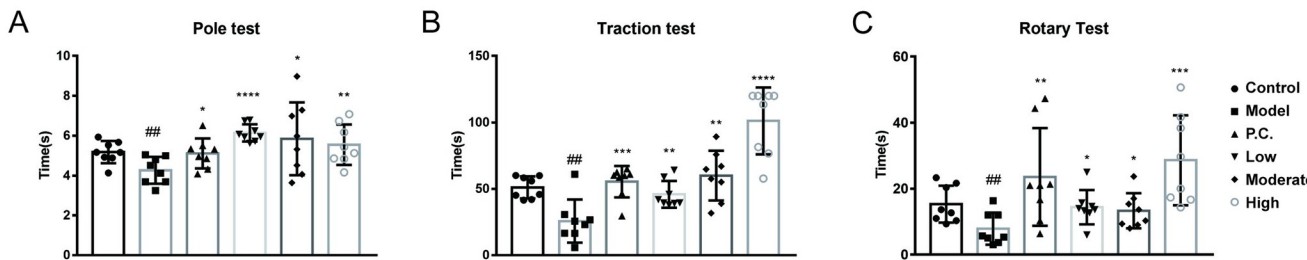

**Fig 2. WEG improves body–coordinated motor dysfunction in PD mice.** (A) Pole test results. (B) Traction test results. (C) Rotary test results. Statistical analyses were performed using one–way ANOVA: *P < 0.05, **P < 0.01, ***P < 0.001, ****P < 0.0001 vs. model group. #P < 0.05, ##P < 0.01 vs. normal group; mean ± SD.

The P.C. group had a longer pole climbing time compared with the model group (P < 0.05). The low group (P < 0.0001), which was the group treated with WEG administration, had the longest pole climbing time. The moderate group (P < 0.05) and the high group (P < 0.01) had prolonged time spent compared with the model group. The PD model mice showed significant dysfunction in limb movement coordination, whereas WEG significantly improved their limb coordination motor dysfunction.

In the traction test (Fig 2B), grasp time was reduced in the model group (P < 0.01) compared with the control group. Grasp time was significantly higher in mice in the high group treated with WEG compared with the model group (P < 0.0001). The P.C. group (P < 0.001), the low group (P < 0.01), and the moderate group (P < 0.01) had significantly higher grip times.

Experimental results from the rotary test showed (Fig 2C) that the dwell time was shorter in the model group compared with the control group (P < 0.01). When compared with the model group, the residence time on the rotating bar was significantly higher in the high group treated with WEG (P < 0.001). Group WEG had improved dwell time compared with P.C. group (P < 0.01), the low group (P < 0.05), and the moderate group (P < 0.05). This result suggested that WEG had the ability to improve the equilibrium disorder in MPTP–induced mice.

Predictors of WEG's ability to improve balance disorders in MPTP–induced mice were indicated by the results of the pole test, rotary test, and traction test.

## Spatial learning and memory skills compared

We used the Morris water maze test to assess the spatial learning and memory abilities of mice. For the positioning test (Fig 3), the time ratios (TR, O/(I+M)) of the sum of swimming time in the outer ring to swimming time in the center and swimming time in the inner ring. In the positioning test (Fig 4), the time to find a platform, the distance swum, TR, and the locomotor thermogram were used to provide feedback on the behavioral ability of the mice. Model mice had significantly higher time to find a plateau (P = 0.0002), distance (P = 0.0005), and TR (P = 0.0075) than the control group. Compared with the model group, treatment with CLCRT and WEG significantly reduced swimming time (P = 0.0003 for P.C. group, P = 0.0004 for the low group, P = 0.0005 for the moderate group, and P = 0.0006 for the high group), reduced distance (P = 0.0029 for P.C. group, P = 0.0016 for the low group, P = 0.0001 for the moderate group, and P = 0.0001 for the high group), and decreased TR values (P = 0.0206 for P.C. group, P = 0.2738 for the low group, P = 0.0111 for the moderate group, and P = 0.0223 for the high group).

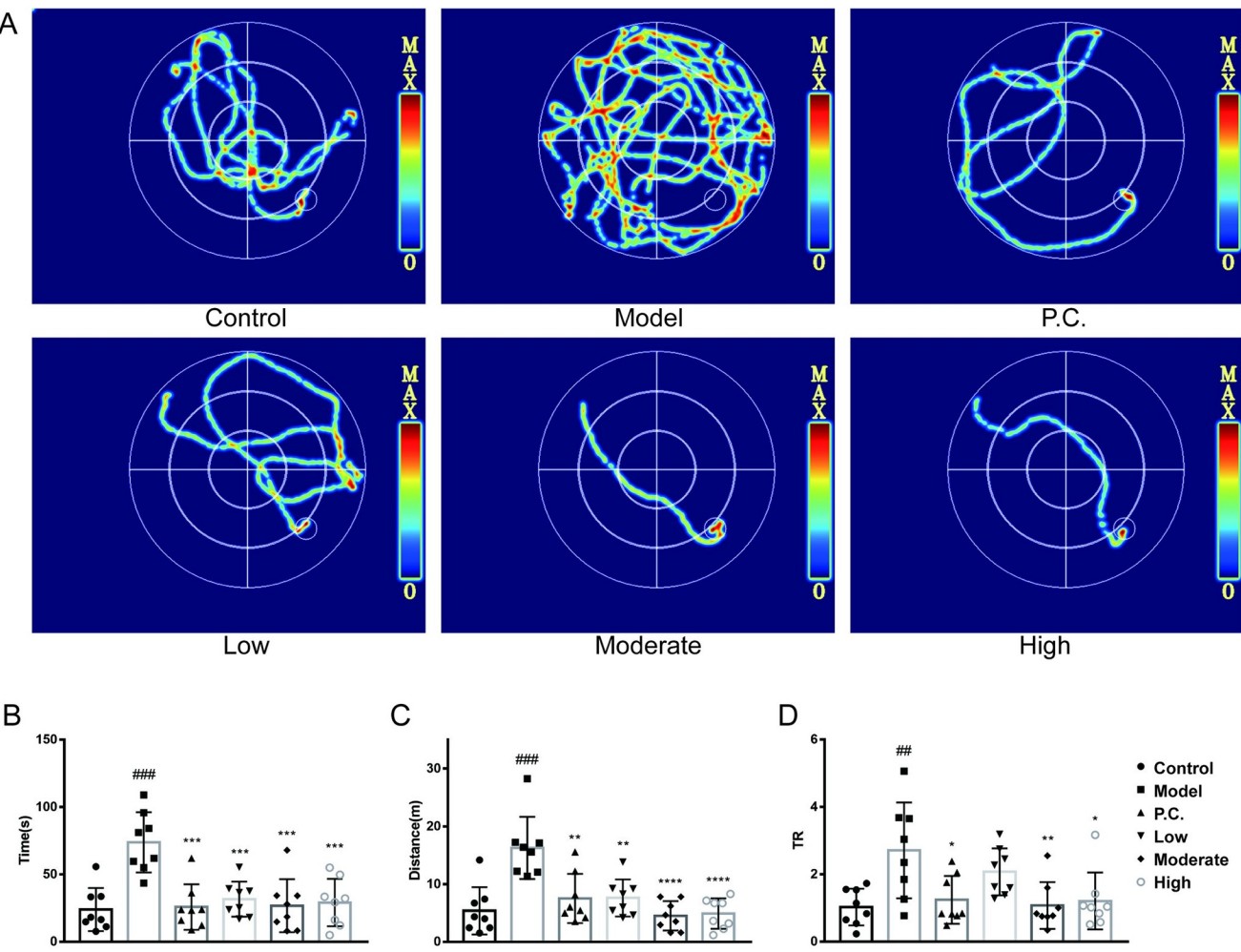

**Fig 3. Effect of WEG on the ability of mice to find platforms in the Morris water maze test.** (A) Trajectory diagram of platform–seeking ability in mice. (B) Time to find the platform (in seconds). (C) Distance of swimming when finding the platform (meters). (D) Ratio of outer circle swimming time to the sum of central and inner circle swimming time TR. Statistical analyses were performed using one–way ANOVA: *$P < 0.05$, **$P < 0.01$, ***$P < 0.001$, ****$P < 0.0001$ vs. model group. #$P < 0.05$, ##$P < 0.01$, ###$P < 0.001$ vs. normal group; mean ± SD.

According to the results of the experiment, the number of passes in the model group ($P = 0.0083$) was significantly less than that of the blank group. Treatment with CLCRT and WEG significantly improved the learning memory ability of the mice. Swimming distance was shorter in the model group compared with the blank group ($P = 0.00002$). TR was greater in the model group ($P = 0.0014$) than in the blank group. There was a greater number of passes over the platform compared with the model group ($P = 0.0486$ for P.C. group, $P = 0.0022$ for the low group, $P = 0.0017$ for the moderate group, and $P = 0.0006$ for the high group). Swimming time was longer compared with the model group ($P = 0.0047$ for P.C. group, $P = 0.0114$ for the low group, $P = 0.0003$ for the moderate group, and $P = 0.0007$ for High). Resting time was shorter compared with the model group ($P = 0.0047$ for P.C. group, $P = 0.0114$ for the low group, $P = 0.0003$ for the moderate group, and $P = 0.0007$ for the high group). Compared with the model group, swimming distance increased ($P = 0.0006$ for P.C. group, $P = 0.0119$ for the low group, $P = 0.0007$ for the moderate group, and $P = 0.0002$ for High). TR values for all groups ($P = 0.0508$ for P.C. group, $P = 0.0332$ the low group, $P = 0.0016$ for the moderate

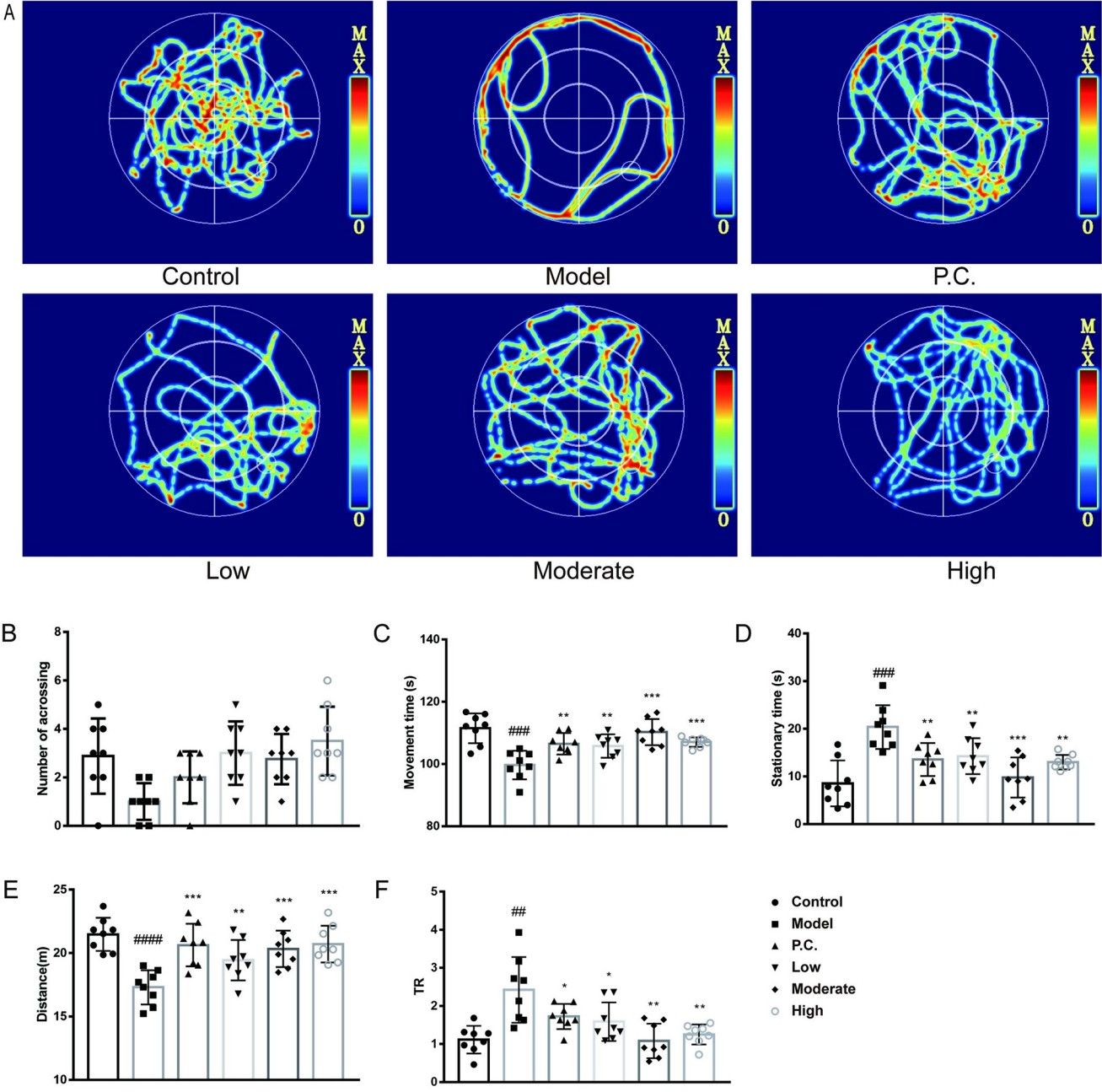

**Fig 4. WEG effects on the probing ability of the Morris water maze test in mice.** (A) Trajectory diagram of mice swimming. (B) Number of times mice passed the platform position. (C) Time spent swimming. (D) Time spent at rest. (E) Distance swimming by mice. (F) TR. Statistical analyses were performed using one–way ANOVA: *P < 0.05, **P < 0.01, ***P < 0.001, ****P < 0.0001 vs. model group. #P < 0.05, ##P < 0.01, ###P < 0.001 vs. normal group; mean ± SD.

group, and P = 0.0025 for the high group) were smaller than the model group. The facts showed that the shortest distance a mouse has to travel to find a platform was across the central two quadrants. From this, we know that the longer the time spent searching for a platform in the outer quadrant among the learning memory of PD mice, the relatively longer the distance a mouse has to travel to find the platform. Thus, the greater the ratio of the time the mice

spent swimming in the outer quadrant compared with the time the mice spent swimming in the inner quadrant, the worse the PD mice were at finding platforms and the worse the spatial memory ability of the mice. Spatial memory ability is an examination of the strength of the mice's brain's ability to learn to remember.

WEG was not concentration–dependent in terms of behavioral improvement in PD mice. Compared with CLCRT, the WEG regimen was more effective in reversing behavioral deficits in PD mice.

## WEG improves brain pathological changes in mice

We stained sections of the hippocampal CA1 area in mice with H&E (Fig 5), and the area ratio of neuronal cells in the CA1 area of the hippocampus and the area ratio of neuronal cells in the striatum were calculated using ImageJ software. The cells in the CA1 area of the hippocampus of the control group were regular in morphology, with plump cell bodies, neat and orderly arrangement, clear nuclei, and intact neuronal morphology (Fig 5A). In the model group, the CA1 area was damaged to some extent, and the nuclei were blurred, as shown by partial loss of neurons, empty stained cytosol, disordered cell arrangement, significantly reduced number of neurons, and incomplete morphology. The improvement in the P.C. group was not significant (Fig 5C). Nerve cells in the striatum of mice in the control group had clear structure and normal morphology with irregular shape (Fig 5B). In the model group, incomplete cell morphology, scattered neuronal cells, reduced number of neuronal cells, and disorganized arrangement were seen in the striatum. More vacuoles were seen throughout the striatum. Compared with the model group, after treatment with WEG, the neuronal cells in the striatum of mice showed different degrees of restoration of morphological structure, and were more densely and neatly arranged. In addition to an increase in the number of cells (Fig 5D), their distribution was more uniform, the cell spaces were reduced, and most of the cells returned to normal morphology.

## Evidence of the effect of WEG on the rate of α–Syn positivity in the substantia nigra (SN) and TH positivity in the striatum of PD mice

α–Syn is a key pathological product of PD pathogenesis. We performed IHC experiments on the substantia nigra of the mouse brain using the rabbit antibody α–Syn (Abcam, ab212184). The results of α–Syn IHC in the SN are shown in Fig 6A, and positive ratios were calculated using ImageJ software (Fig 6C). α–Syn–positive cells were seen less frequently in the substantia nigra of the control mice, mainly in the cytoplasm and cytosol. We found an increase in α–Syn positive cells in the substantia nigra in the model group compared with the control group ($P < 0.0000$). CLCRT and WEG reduced the rate of α–Syn positivity in the substantia nigra. WEG improved the aggregation of α–Syn in the brain of MPTP–induced PD mice. We performed assays for striatal IHC of mice brains using the rabbit antibody TH (Abcam, ab137869), and positive ratios were calculated using ImageJ software (Fig 6B and 6D). In the striate tissue of the control group, TH–positive cells were more abundant, mainly in the cytoplasm and cytosol, than in the blank group ($p < 0.0000$). Striatal TH positivity was increased in the model group compared with the striatum ($P < 0.0000$ for all groups).

## Effect of WEG on the expression levels of TH and α–Syn protein in the brain of PD mice

With a view of exploring the treatment action of WEG on MPTP–induced PD in mice, we assessed TH and α–Syn activation according to western blot (Fig 7). The MPTP treated mice in the model group showed significantly reduced expression of TH and significantly increased

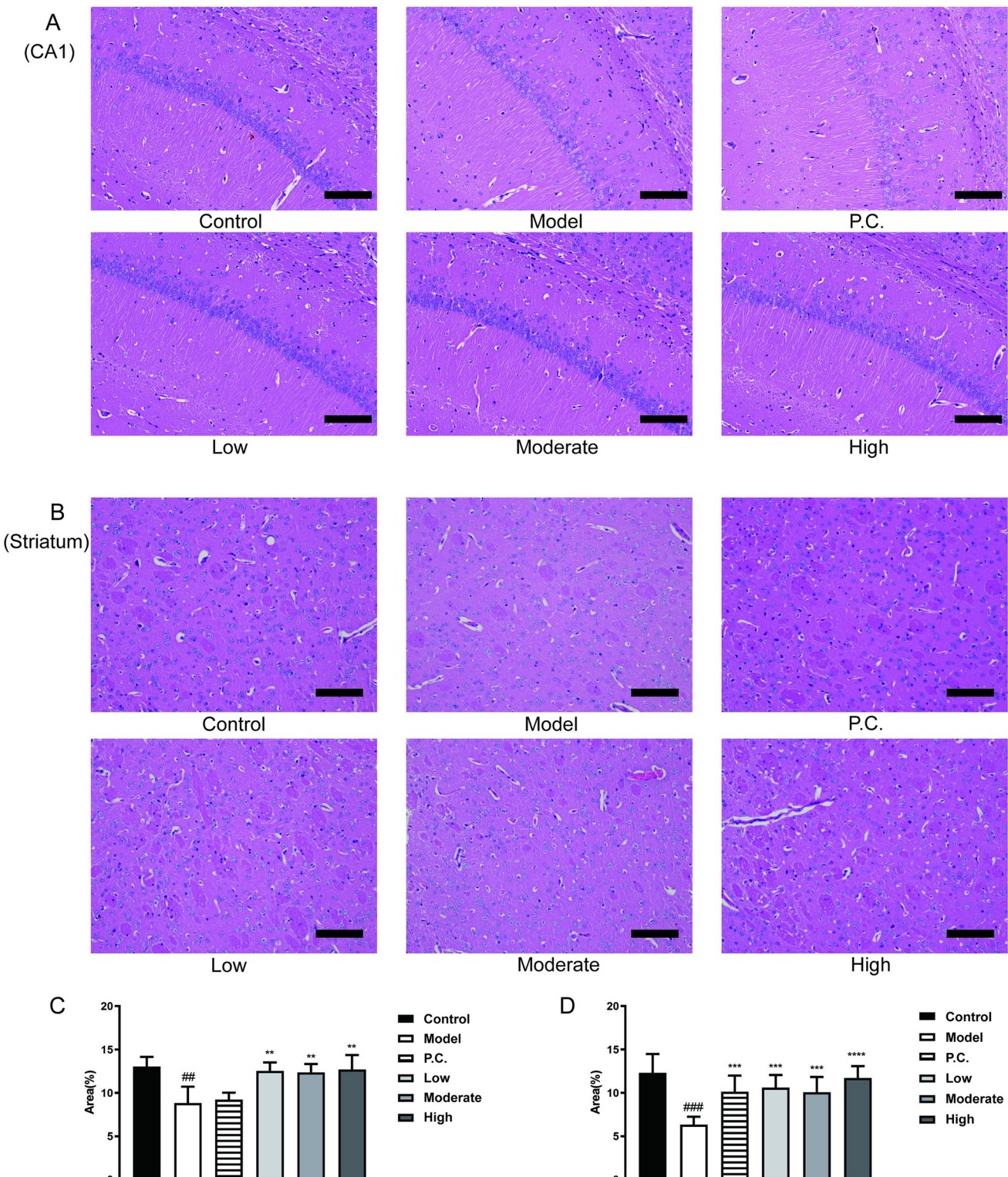

**Fig 5. WEG effects on MPTP–induced pathological changes in the hippocampal CA1 region and striatum of mice.** (A) H&E staining of the hippocampal CA1 region of mice (×200, 100 μm). (B) H&E staining of the striatum of mice (×200, 100 μm). (C) Percentage of neuronal cells in CA1 region of hippocampus (%). (D) Percentage of neuronal cells in the striatum (%). Statistical analyses were performed using one–way ANOVA: *$P < 0.05$ **$P < 0.01$, ***$P < 0.001$, ****$P < 0.0001$ vs. model group. #$P < 0.05$, ##$P < 0.01$ vs. normal group; mean ± SD.

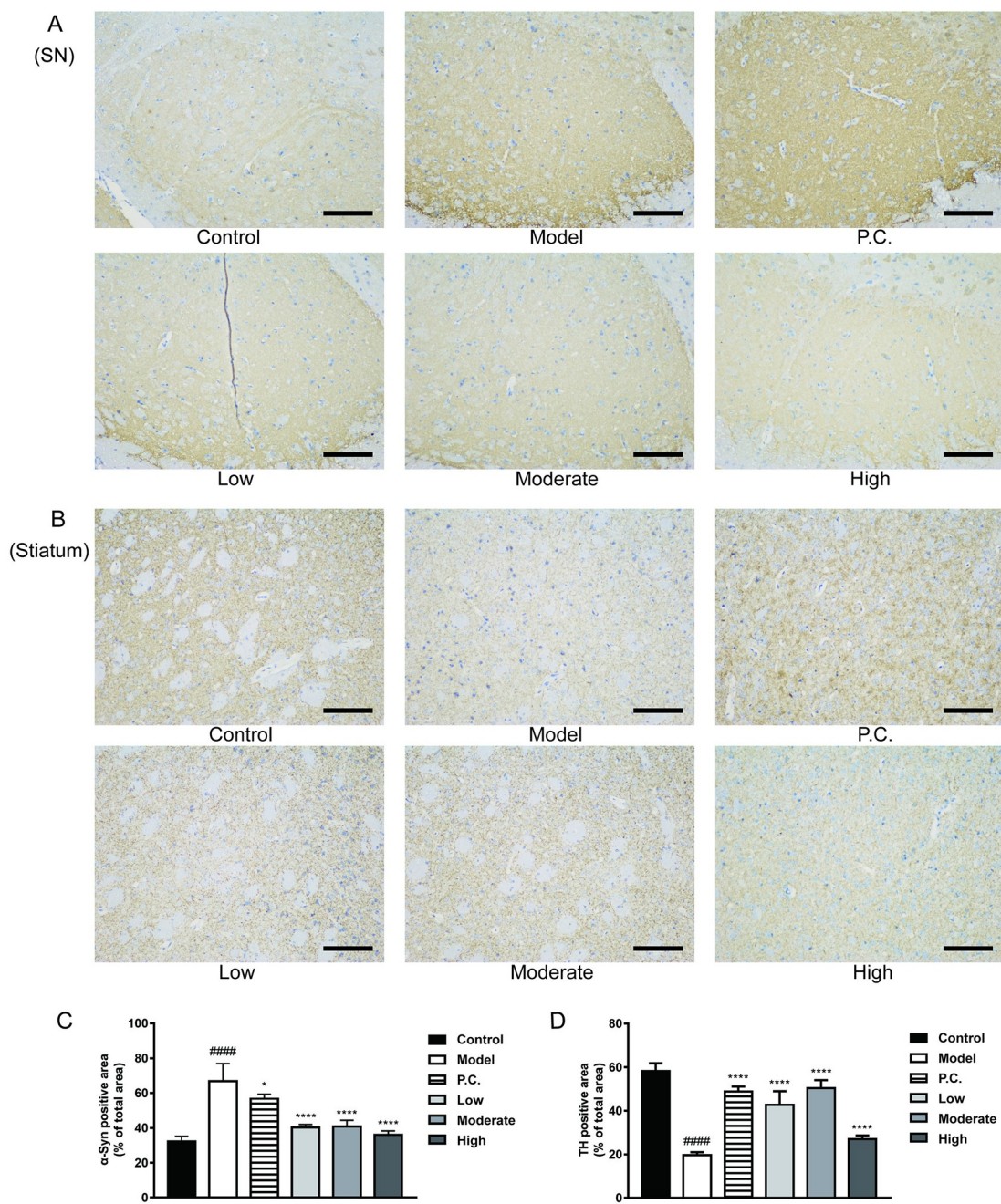

**Fig 6. WEG results of IHC experiments in the brain of PD mice.** (A) IHC staining of WEG for positive expression of α–syn in the SN of PD mice (×200, 100 μm). (B) IHC staining of WEG for positive expression of TH in the striatum of PD model mice (×200, 100 μm). (C) Percentage of area positive for α–syn in each group (%). (D) Percentage of area positive for TH in each group (%). Statistical analyses were performed using one–way ANOVA: *$P < 0.05$, **$P < 0.01$, ***$P < 0.001$, ****$P < 0.0001$ vs. model group. #$P < 0.05$, ##$P < 0.01$, ###$P < 0.001$, ####$P < 0.0001$ vs. normal group; mean ± SD.

expression of α–Syn in brain tissue compared with the control group. The expression level of TH protein was up–regulated and that of α–Syn protein was significantly down–regulated in the brain tissue of mice after WEG intervention. WEG had a neuroprotective effect on MPTP–induced PD mice and improved the expression levels of TH and α–Syn in the brain.

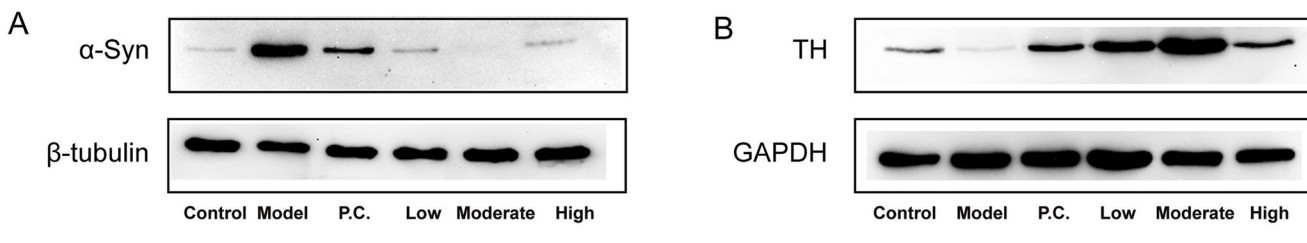

**Fig 7. Effect of WEG therapy on expression of (A) α–Syn and (B) TH.**

## WEG improves pathological changes in the small intestine

We performed H&E staining (Fig 8A) and periodic acid–Schiff (PAS) staining (Fig 8B) on sections of mouse small intestine, and calculated the length of the small intestinal villus (μm) as well as the area ratio of goblet cells (%) using ImageJ software. The intestinal mucosa of the small intestine of the blank group of mice was pink, with intact and regular villus structure and neat margins; the crypt was relatively shallow, the small intestinal glands were intact, and the number of goblet cells was high (PAS treatment showed positive results). Compared with the control group, the model group showed broken or detached small intestinal villus (P = 0.0000), a significant decrease in the number of goblet cells (P = 0.0000), significant destruction of the villus structure, and more severe damage to the small intestinal glands. WEG had an improvement effect on intestinal tissue in PD model mice. Compared with the model group, the WEG–treated mice showed longer villus length (P = 0.0414 for the low group, P = 0.0040 for the moderate group, and P = 0.0004 for the high group) and a higher number of goblet cells (P = 0.0017 for the low group, P = 0.0149 for the moderate group, and P = 0.0004 for the high group). (P = 0.0149 for the moderate group and P = 0.0001 for the high group.) The reduced depth of the crypt was associated with improved damage to the intestinal mucosa and small intestinal glands and a corresponding increase in the digestion and absorption of nutrients.

## WEG improves gut microbiota dysbiosis in PD mice

In this study, the effects of different doses of WEG on the gut microbiota of PD mice were observed from different units. According to structural analysis of the microbial colonies at the phylum unit for comparison among groups, the following seven phyla were common to all samples: *Firmicutes*, *Bacteroidetes*, *Proteobacteria*, *Saccharibacteria* (*TM7*), *Actinobacteria*, *Deferribacteres*, and *Tenericutes*. Normal mice gut microbiota consisted mainly of *Firmicutes* and *Bacteroidetes*, with the sum of the two accounting for more than 97% of the total microbiota. The relative abundance ratios of *Firmicutes* and *Bacteroidetes* were analyzed comparatively. The model group had less *Firmicutes* (P < 0.0000) and more *Bacteroidetes* (P = 0.0001) compared with the control group. Compared with the model group, the WEG resulted in an increased proportion of *Firmicutes* (P = 0.0059 for the low group, P = 0.0365 for the moderate group, and P = 0.0124 for the high group) and a decreased proportion of *Bacteroidetes* in the mice intestine (P = 0.0026 for the low group, P = 0.0347 for the moderate group, and P = 0.0090 for the high group). In addition, CLCRT caused an increase in the proportion of *Firmicutes* (P = 0.0001) and a decrease in the proportion of *Bacteroidetes* (P = 0.0002) in the intestine of mice.

According to the microbial colony structure analysis at the order level, the normal mice gut microbiota consisted mainly of the orders *Lactobacillales*, *Clostridiales*, and *Bacteroidales*,

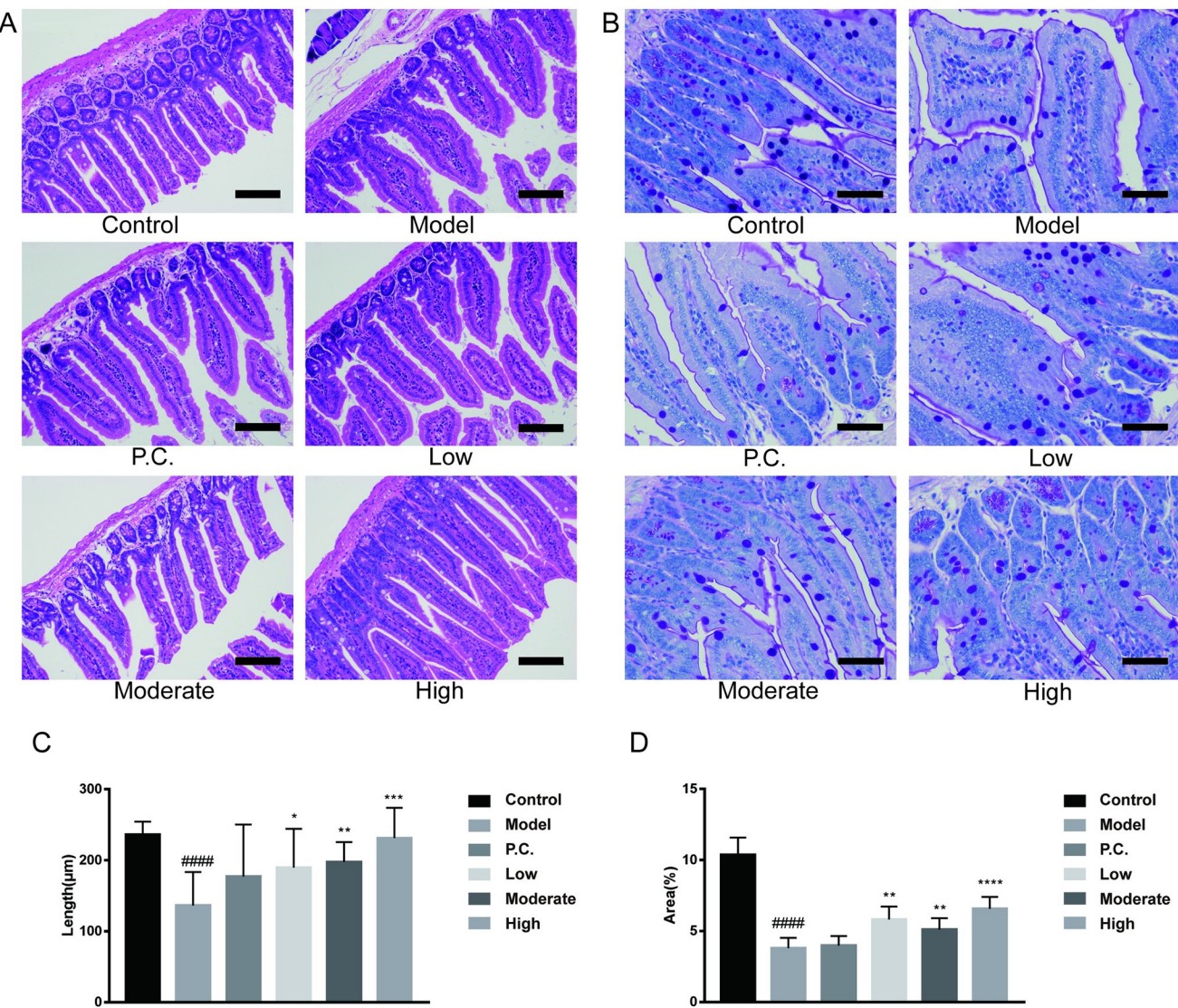

**Fig 8. Effect of WEG on MPTP-induced histopathological changes in the small intestine of mice.** The small intestine of mice was stained with H&E and PAS to observe the morphological changes in the small intestine. (A) H&E staining of small intestine (×200). (B) PAS staining of small intestine (×400). (C) Length of small intestine villi in each group (μm). (D) Positive area in each group of sections (%). Statistical analyses were performed using one-way ANOVA: *P < 0.05, **P < 0.01, ***P < 0.001, ****P < 0.0001 vs. model group. #P < 0.05, ##P < 0.01, ###P < 0.001 ####P < 0.0001 vs. normal group, mean ± SD.

accounting for more than 97% of the total number of bacteria. *Lactobacillales* were reduced in the model group compared with the control group (P = 0.0016). Both CLCRT and WEG increased the amount of *Lactobacillales* in the intestine of mice compared with the model group (P = 0.0038 for the P.C. group, P = 0.0001 for the low group, P = 0.0306 for the moderate group, and P = 0.0044 for the high group). *Bacteroidales* were less abundant in the model group compared with the control group (P = 0.0055). Both CLCRT and high doses of WEG elevated the amount of *Clostridiales* in the intestine of mice compared with the model group (P = 0.0304 for the P.C. group and P = 0.0509 for the high group), but the low and moderate doses of WEG did not have a significant effect on *Clostridiales* in the intestine of mice.

*Bacteroidales* were reduced in the *Modelmodel* group compared with the control group (P = 0.0001). Both CLCRT and WEG resulted in increased levels of *Bacteroidales* in the intestine of mice compared with the model group (P = 0.0002 for the P.C. group, P = 0.0026 for the low group, P = 0.0347 for the moderate group, and P = 0.0071 for the high group).

According to the genus unit microbial colony structure analysis, the normal mice gut microbiota consisted mainly of *Lactobacillus*, *Oscillospira*, and *Streptococcus*, accounting for more than 50% of the total microbiota. *Lactobacillus* levels were reduced in the model group compared with the control group (P = 0.0052) and *Streptococcus* levels were also reduced in the model group (P = 0.0054). Both CLCRT and WEG increased *Streptococcus* levels in the intestine of mice compared with the model group (P = 0.0107 for the P.C. group, P = 0.0001 for the low group, and 0.0102 for high group). Ginseng showed significant improvement in gut microbiota disorders in PD mice after a low-dose WEG administration, which most closely resembled the gut flora composition of normal mice. In addition, *Oscillospira* was not significantly different between the groups, although *Oscillospira* accounted for approximately 3% of the gut microbiota of the mice. MPTP-induced PD was not associated with *Oscillospira*, however, and we did not find any significant difference between the groups. The WEG intervention significantly increased the levels of beneficial bacteria, such as *Lactobacillus*. Treatment with WEG restored the abundance of probiotic bacteria and reduced the levels of harmful microbiota.

We drew a clustering heat map based on the genus unit, reflecting the trend of species abundance distribution among the groups, with red representing positive correlation and blue representing negative correlation (Fig 9D). The figure clearly shows that the model group clustered separately, whereas the remaining five treatment groups clustered into one category with the normal group. Among them, WEG not only interfered with the elevation of beneficial genera, such as Lactobacillus, but also made the elevation of other beneficial genera related to glycolipid metabolism, such as *Allobaculum* and *Adlercreutzia*, which improved the flora environment in the body.

High doses of WEG did not reduce the level of *Odoribacter* and *Enterococcus* and other genera in the intestine, but rather elevated them. Therefore, although a high dose of WEG could improve PD, it also affected the intestinal bacteria in mice. As shown by the principal component score plot, the WEG and CLCRT treatment groups were relatively concentrated with the blank group in terms of dispersion, and the model group had less overlap with the rest of the groups. Additionally, the first principal component in this principal component analysis accounted for 74.7%, and the second principal component accounted for 24.7%, with a cumulative contribution value of 99.4%, reaching more than 85% of the eigenvalues (Fig 9E).

We conducted an evaluation of community richness and diversity in the gut of mice using the alpha-diversity index. A rarefaction curve was plotted using the index type Chao1 index (Fig 9F). When the extraction depth reached above 20,000, the rarefaction curve leveled off, indicating that the amount of sequencing was sufficient to cover all taxa in the sample. For a more comprehensive assessment of the alpha diversity of the microbial community, we used an index such as Chao1 (Fig 10A). In the alpha-diversity analysis, the indices for each WEG dosing group were close to the control group with significant differences compared with the model group (p < 0.05 for all group). The results of the alpha analysis indicated that treatment with WEG normalized the diversity and richness of the gut microbiota. Beta diversity was analyzed using both principal coordinates analysis (PCoA) and nonmeasured multidimensional scale analysis (NMDS) to describe the variability of the samples (group to group). We evaluated the distance of similarity between samples by NMDS analysis (Fig 10B). The distances between the model group and the other groups were clearly separated, while the WEG treatment group was close to the control group. The results indicated that treatment with WEG led

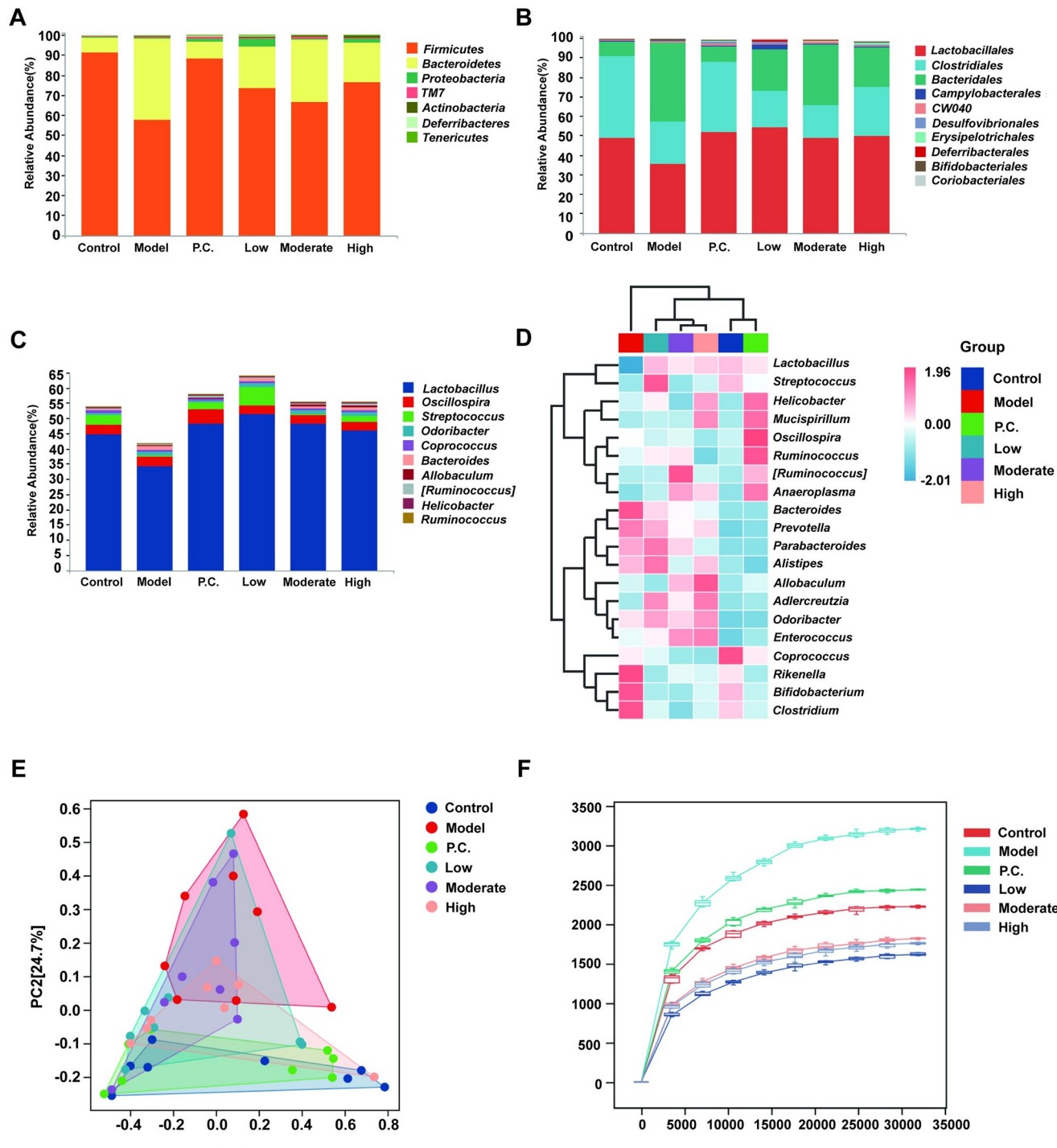

**Fig 9. Effect of WEG in different basic units on the composition of gut microbiota in PD mice.** (A) Effect of WEG based on order species unit on gut microbiota in PD mice. (B) Effect of WEG based on phylum species unit on gut microbiota in PD mice. (C) Effect of WEG based on genus species unit on gut microbiota in PD mice. (D) Heat map for genus unit based. (E) Principal component analysis. (F) Rarefaction curves based on Chao1 index.

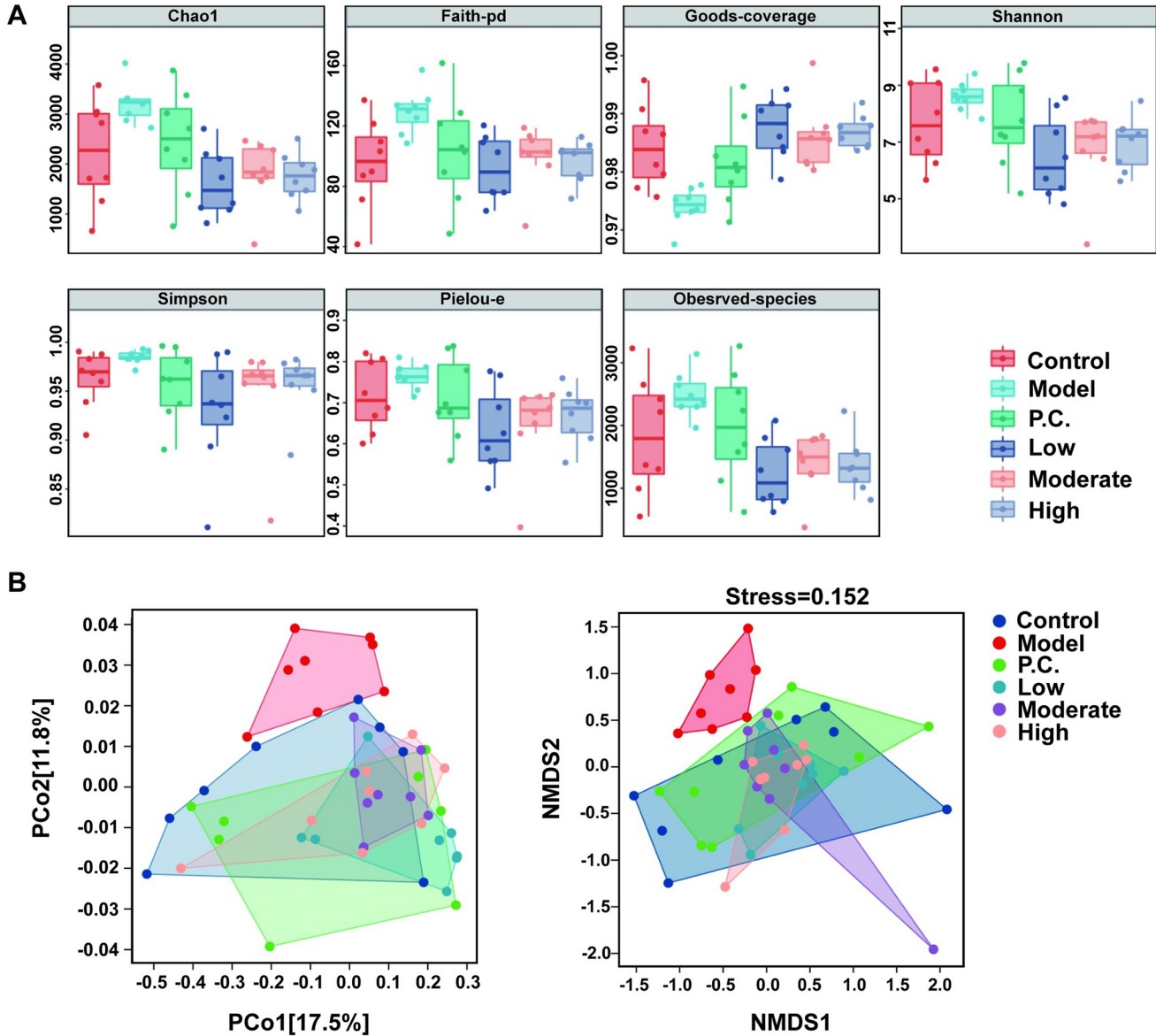

**Fig 10. Effect of WEG on the abundance and diversity of gut microbiota in PD mice.** (A) Results of the alpha-diversity evaluation according to each index. (B) PCoA and NMDS results of beta-diversity.

to a convergence of intergroup variability in the gut microbiota of PD mice with the normal group.

## Discussion

Ginseng has complex active components, and the chemical composition of ginseng extracts obtained under different processing conditions varies greatly [37]. The choice of temperature and solvent influenced the ratio of various chemical components in the extract of ginseng [38]. Kim et al. found that the content of protopanaxadiol (PPD)-type ginsenosides in ginseng decreased with increasing temperature, whereas the content of protopanaxatriol (PPT)-type ginsenosides increased after high temperature conversion, and the content of some

ginsenosides, such as ginsenoside Rf, decreased with increasing temperature [39–41]. The conventional method of high-temperature extraction of ginseng broke down the proteins in ginseng, resulting in a near-absent protein content in the resulting sample [42–44]. The contents of polysaccharides, ginsenosides, and proteins in ginseng were 40%, 4%, and 10%, respectively [45–47]. In this study, the contents of ginseng polysaccharides, ginsenosides, and proteins in WEG obtained using mild extraction conditions of low temperature and water extraction were 26.58%, 3.85%, and 7.36%, respectively. Our study was statistically evaluated with behavioral and histopathological data from mice. We found that WEG had a significant ameliorative effect on MPTP-induced PD mice, based on the pharmacological effects exerted by the joint coordination between the three main compounds in WEG. For MPTP-induced impairment of brain α-Syn accumulation in mice, our study found that oral administration of WEG was sufficient to reduce brain α-Syn accumulation and increase TH content in PD mice.

MPTP freely crosses the blood–brain barrier and is converted to $MPP^+$ in the brain by astrocytes. $MPP^+$ is selectively taken up by dopaminergic cells and inhibits mitochondrial complexes in the respiratory chain [48]. MPTP toxicity-induced PD models are well-established animal models, and numerous studies have shown that MPTP can cause impaired learning memory in humans, mice, and primates [49]. MPTP is the only environmental factor directly associated with the development of levodopa-responsive PD, which is clinically indistinguishable from PD. Similar chemicals abound, however, supporting the hypothesis that substances in the environment may contribute to this disorder [50]. Some scholars found that ginsenoside $Rb_1$ could alleviate MPTP-induced gait kinetic dysfunction and cognitive impairment in mice, suggesting that ginsenoside $Rb_1$ has neuroprotective mechanisms in PD mice [51]. We performed pathological studies on the hippocampal CA1 and striatum of MPTP-injured mice and the small intestine of mice and discovered that damaged neuronal cells in the CA1 and striatum of WEG-treated mice were improved, and the number of neuronal cells increased and the cell morphology largely returned to normal after treatment with WEG. Furthermore, WEG also had a significant therapeutic effect on the damaged mice intestine, with improvements in the damaged intestinal mucosa and small intestinal glands, and a corresponding increase in the digestion and absorption of nutrients.

People with PD suffer from clinical manifestations of tremor, muscle tonus, motor bradykinesia, and postural dysreflexia. Its main pathological manifestations are pathophysiological loss or degeneration of dopaminergic neurons in the midbrain substantia nigra and the development of neuronal Lewy bodies with reduced striatal dopamine levels. Moreover, PD is based on the brain, so the examination of learning and memory abilities is also the focus of behavioral tests in mice [52–56]. Success in simulating working memory deficits and other cognitive impairments in PD also have been reported for the MPTP model [57–60]. First, we used behavioral tests, such as the pole test, to determine the strength of the limbs as a basis for the presence of motor deficits in mice. MPTP altered the locomotor ability of the mice, suggesting that these mice had a locomotor deficit, whereas treatment with WEG improved the mice's locomotor impairment well, which was consistent with our previously hypothesized results. Notably, the results of the Morris water maze test provided a more intuitive sense that WEG could improve motor retardation and learning memory abilities in mice. These results support the finding that WEG can improve spatial learning and cognitive abilities in mice already disturbed by MPTP in in vivo experiments. α-Syn, a key protein in PD development, and Lewy bodies formed by α-Syn aggregation are the main pathological features of PD [61]. Min et al. proposed that gintonin, a glycolipid protein isolated from ginseng, has some neuroprotective effects. Gintonin's neuroprotective effects were associated with reduced accumulation of α-Syn in the substantia nigra and striatum of MPTP-induced PD mice, and the neuroprotective effect of gintonin was further validated via analyzing the effect of gintonin on $MPP^+$ treated

SH-SY5Y cells [62]. This result is consistent with the findings in this study that WEG intervention reduced the accumulation of α-Syn and thus acted as a neuroprotective agent. This finding also revealed that WEG has played a role in resisting the progression of PD, and the results of both WB and IHC experiments suggested that WEG clearance of α-Syn appears to be more effective than CLCRT given alone. Additionally, it is important to measure the TH levels in the brain. TH catalyzes the rate-limiting step in the synthesis of the neurotransmitter dopamine and other catecholamines. TH enzymes play an important role in dopamine synthesis and are abundant in dopaminergic neurons. Therefore, TH is used as a marker of dopaminergic neurons and plays an important role in the pathophysiology of PD [63]. Establishing the beneficial effect of gintonin on MPTP-induced nigrostriatal TH loss in previous reports has confirmed the hypothesis that gintonin can restore reduced TH expression in MPTP-treated mice [64]. We found that the MPTP-induced decrease in TH levels in the mouse brain and the intervention of WEG modulated the increase in intracranial TH levels, thus preventing the loss of TH in the skull and acting as a neuroprotective agent.

Early studies have found that the etiology of PD may originate in the gut [65]. Researchers detected α-Syn in the intestinal mucosa and submucosal nerve fibers of PD patients, and found that α-Syn in the gut could be transported to the brain through the vagus nerve, thus providing a basis for the hypothesis that the origin of PD may come from outside the brain in the gut [66]. Changes in the composition of the gut microbiota also have been observed in the intestines of patients with PD [67]. When mice were cocolonized with the flora of PD patients by the fecal microbiota transplantation technique, the mice exhibited typical signs of PD such as motor deficits and neuroinflammation [68]. Previous studies have reported that MPTP induction can increase the number of some pathogens and have confirmed that MPTP neurotoxicity to the brain can lead to intestinal dysfunction and ultimately alter gut microbiota in PD patients [69, 70]. This study confirmed that MPTP did have a detrimental effect on the intestine and gut microbiota of mice, whereas WEG not only improved the intestinal tissues, but also had the ability to improve the disordered flora of mice. WEG increased the proportion of *Firmicutes* and decreased the proportion of *Bacteroidetes* in the intestine of mice and repaired the disordered gut microbiota, resulting in a normal regulation of the active system of the flora. The gut microbiota is complex in structure and consists of four main bacterial phyla. The thick-walled phylum is predominant (50–75%), followed by the phylum *Bacteroides* (10–50%), the phylum *Actinomycetes*, and the phylum *Aspergillus* [71]. Elevated flora of either phylum will affect the levels of other species and lead to disorders. We found a decrease in *Lactobacillus* abundance in the model group and a significant increase with oral WEG, and other studied also have confirmed that elevated values of *Lactobacillus* abundance are important for improving the gut microbiota environment [72, 73]. The high dose of WEG in this study improved PD and also had an effect on the gut microbiota of the mice. Even though the high dose of WEG had the best effect on the repair of the intestinal environment, the treatment was better for the low dose of WEG. Thus, the hypothesis that an imbalance of gut microbiota may indirectly contribute to the pathogenesis of PD is further supported by the fact that the mechanism of change in gut microbiota regulation by WEG. Whether or not WEG can be used in combination with CLCRT needs to be further explored. Also, our study has not investigated which metabolic pathways are involved in the neuroprotective mechanism of WEG improvement in PD mice, and this pathway should be investigated in subsequent experiments.

## Conclusion

This study used a low-temperature extraction technique to obtain WEG with high retention of active ingredients, and MPTP intraperitoneal injection was used to construct an in vivo PD

model. We demonstrated that WEG could repair intestinal damage and improve gut microbiota dysbiosis in PD mice. We also demonstrated that WEG could significantly improve behavioral functions and inhibit the accumulation of α-Syn in the brain via regulating the level of TH in the brain, thus exerting neuroprotective effects in PD mice. The study provides a new approach to the pharmacological treatment of PD, further supports the hypothesis that the pathogenesis of PD is related to the gut, and offers a research basis for whether WEG may be a suitable treatment for other degenerative diseases.

## Supporting information

**S1 Raw images.**
(PDF)

## Author Contributions

**Conceptualization:** Yu Zhao.

**Data curation:** Jie Li, Bo Pang.

**Formal analysis:** Ning Xu, Jie Li, Yu Zhao.

**Funding acquisition:** Yu Zhao.

**Investigation:** Ning Xu, Yu Zhao.

**Methodology:** Ning Xu, Shuyang Xing, Bo Pang, Yu Zhao.

**Project administration:** Jie Li, Meichen Liu, Meiling Fan, Yu Zhao.

**Resources:** Meichen Liu, Meiling Fan, Yu Zhao.

**Software:** Ning Xu.

**Supervision:** Yu Zhao.

**Visualization:** Ning Xu.

**Writing – original draft:** Ning Xu, Yu Zhao.

**Writing – review & editing:** Ning Xu, Shuyang Xing.

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
