## [Decision Letter · Decision Letter 0]

2 Jun 2023

PONE-D-23-02610

Water extract of ginseng alleviates parkinsonism in MPTP–induced Parkinson's disease mice

PLOS ONE

Dear Dr. Zhao,

Thank you for submitting your manuscript to PLOS ONE. After careful consideration, we feel that it has merit but does not fully meet PLOS ONE’s publication criteria as it currently stands. Therefore, we invite you to submit a revised version of the manuscript that addresses the points raised during the review process.

We look forward to receiving your revised manuscript.

Kind regards,

Chun-Hua Wang

Academic Editor

PLOS ONE

Journal Requirements:

"This work was supported by Erasmus+ project My Home - My Science Lab. We are grateful to the members of My Home - My Science Lab project and teachers involved in the study."

6. Your striking image file will represent your article upon publication on the PLOS ONE homepage. The image must be derived from a figure or supporting information file from your manuscript. Ideally, striking images should be high resolution, eye-catching, single panel images that do no contain additional text, scale bars, or arrows.

Please also keep in mind that PLOS's Creative Commons Attribution License applies to striking images. As such, please do not submit any figures or photos that have been previously copyrighted unless you have express written permission from the copyright holder to publish under the CCAL license. You can read more about PLOS’s Creative Commons License on our homepage: http://journals.plos.org/plosone/s/licenses-and-copyright

Reviewers' comments:

Reviewer's Responses to Questions

**Comments to the Author**

1. Is the manuscript technically sound, and do the data support the conclusions?

Reviewer #1: Yes

Reviewer #2: Partly

Reviewer #3: Yes

2. Has the statistical analysis been performed appropriately and rigorously? 

Reviewer #1: Yes

Reviewer #2: Yes

Reviewer #3: Yes

3. Have the authors made all data underlying the findings in their manuscript fully available?

Reviewer #1: Yes

Reviewer #2: Yes

Reviewer #3: Yes

4. Is the manuscript presented in an intelligible fashion and written in standard English?

Reviewer #1: Yes

Reviewer #2: Yes

Reviewer #3: Yes

5. Review Comments to the Author

Reviewer #1: In this study the authors used a low-temperature extraction technique to obtain WEG with high retention of active ingredients. The authors demonstrated that WEG could repair intestinal damage and improve gut microbiota dysbiosis in PD mice. The authors also demonstrated that WEG could significantly improve behavioral functions and inhibit the accumulation of α-Syn in the brain via regulating the level of TH in the brain, thus exerting neuroprotective effects in PD mice.

This study provides a new approach to the pharmacological treatment of PD, further supports the hypothesis that the pathogenesis of PD is related to the gut, and offers a research basis for whether WEG may be a suitable treatment for other degenerative diseases.

Reviewer #2: In this study, they explored the neuroprotective effects of water extract of ginseng (WEG) on the brains of Parkinson's disease (PD) mice and its ability to improve their damaged intestinal system, addressing dyskinesia. Their data showed that WEG protected damaged neuronal cells, inhibited α-synuclein aggregation, and increased tyrosine hydroxylase expression in the mice's brains. WEG also significantly improved intestinal damage, regulated gut disorders, increased beneficial bacteria like Lactobacillus, and restored bacterial abundance and diversity. Overall, WEG protected neurons in PD mice's brains, regulated gut microbiota, improved behavioral disorders, and offered therapeutic effects for PD mice.

The article has some innovativeness but has the following issues:

Main comment:

1. The authors included an analysis of the gut microbiome as an indicator when exploring the neuroprotective effects of WEG on PD, and indeed, the relationship between gut microbiome and PD is a new and broad research direction. However, the authors did not explicitly state the reason for including the gut microbiome analysis in the introduction. In fact, this indicator makes the article appear loose and unfocused. It is confusing whether WEG protects neurons because of its effects on the gut microbiome, or if it affects the gut microbiome because of its neuroprotective properties. It would be better to directly analyze the mechanisms affecting neurons instead.

Other comments:

1. What are the specific doses of WEG in the low, middle, and high groups in the article, and what is the basis for these doses? The article mentions oral administration of WEG from day 28 to 56. How was this application period determined? In pharmacological experiments, the establishment of dosage and timing requires sufficient justification.

2. In Figure 6, the IHC staining of WEG is shown at x400 magnification, which is quite large. Are there images available at x100 or x50 magnification? This would provide clearer comparisons.

3. Please consider streamlining the article, particularly the introduction and discussion sections. Any redundant content can be removed.

Reviewer #3: The study focused on the neuroprotective effects of water extract of ginseng (WEG) on a mouse model of Parkinson's Disease (PD) induced by MPTP (1-methyl-4-phenyl-1,2,3,6-tetrahydropyridine). The authors presented significant results indicating that WEG could ameliorate MPTP-induced PD-like symptoms, improve intestinal damage, reduce α-Syn accumulation, increase TH level, and rectify gut microbiota dysbiosis. The study provides valuable insights into the potential of WEG as a pharmacological treatment for PD. However, several points could be further clarified or improved.

1) The authors mentioned that WEG extraction was performed under mild conditions of low temperature and water extraction. It would be interesting to compare these results with those obtained under different conditions. More discussion on the extraction technique might enhance the understanding of the effects of extraction conditions on the efficacy of WEG.

2)Although the authors suggest that WEG plays a role in resisting PD progression, the exact mechanisms by which WEG exerts its effects are not completely elucidated. Further exploration of this could provide a more detailed understanding of the mechanisms underlying the effects of WEG.

3)It was mentioned that the high dose of WEG improved PD and also affected the gut microbiota of the mice. More information or discussion on the correlation between dosage and effects would be beneficial.

4) For figure 5 and figure 6, a low magnification overview is needed for the structure of the brain to ensure the zoom in image was taken from the same location.

5) For figure 7B, what's the top band of GAPDH? GAPDH should only have one band.

6. PLOS authors have the option to publish the peer review history of their article (what does this mean?). If published, this will include your full peer review and any attached files.

Reviewer #1: No

Reviewer #2: No

Reviewer #3: No

---

## [Author Response · Author response to Decision Letter 0]

2 Aug 2023

Reviewer Comments:

Reviewer 1

In this study the authors used a low-temperature extraction technique to obtain WEG with high retention of active ingredients. The authors demonstrated that WEG could repair intestinal damage and improve gut microbiota dysbiosis in PD mice. The authors also demonstrated that WEG could significantly improve behavioral functions and inhibit the accumulation of α-Syn in the brain via regulating the level of TH in the brain, thus exerting neuroprotective effects in PD mice.

This study provides a new approach to the pharmacological treatment of PD, further supports the hypothesis that the pathogenesis of PD is related to the gut, and offers a research basis for whether WEG may be a suitable treatment for other degenerative diseases.

Reviewer 2

In this study, they explored the neuroprotective effects of water extract of ginseng (WEG) on the brains of Parkinson's disease (PD) mice and its ability to improve their damaged intestinal system, addressing dyskinesia. Their data showed that WEG protected damaged neuronal cells, inhibited α-synuclein aggregation, and increased tyrosine hydroxylase expression in the mice's brains. WEG also significantly improved intestinal damage, regulated gut disorders, increased beneficial bacteria like Lactobacillus, and restored bacterial abundance and diversity. Overall, WEG protected neurons in PD mice's brains, regulated gut microbiota, improved behavioral disorders, and offered therapeutic effects for PD mice.

The article has some innovativeness but has the following issues:

Comment 1:

The authors included an analysis of the gut microbiome as an indicator when exploring the neuroprotective effects of WEG on PD, and indeed, the relationship between gut microbiome and PD is a new and broad research direction. However, the authors did not explicitly state the reason for including the gut microbiome analysis in the introduction. In fact, this indicator makes the article appear loose and unfocused. It is confusing whether WEG protects neurons because of its effects on the gut microbiome, or if it affects the gut microbiome because of its neuroprotective properties. It would be better to directly analyze the mechanisms affecting neurons instead.

Authors’ responses:

We have included the following in the introduction by reviewing references [10-12] and [31-32].

The latest research has found that PD may originate in the intestines. Since abnormal α-Syn first appears in intestinal neurons before the onset of PD, intestinal dysbacteriosis is most likely related to the onset of PD disease [10]. Intestinal microorganisms act on the central nervous system through metabolites, neurotransmitters and cytokines secreted by immune cells, regulating brain-gut axis interactions [11], and the imbalance of the gut microbiota of Parkinson's patients may lead to a-Syn misfolding, from the intestinal nerves to the brain and brainstem, which in turn leads to brain dopaminergic neuronal damage [12].

At the same time, ginsenosides also have a potential therapeutic effect on intestinal microbiota dysbiosis. Ginsenoside Rb1 significantly changed the composition of gut microbiota in hyperglycemic mice and increased the abundance of mucin-degrading bacteria Akkermansia spp. [31]. Ginsenoside Rk3 intake caused significant changes in the intestinal flora of mice, enriching Bacteroides, Alloprevotella and Blautia, significantly reducing the Firmicutes/Bacteroidetes ratio, effectively improving intestinal dysbacteriosis and reducing the inflammatory response of mice [32].

[10] Zhang Y, Xu S, Qian Y. (2023). Sodium butyrate ameliorates gut dysfunction and motor deficits in a mouse model of Parkinson’s disease by regulating gut microbiota [J]. Frontiers in Aging Neuroscience, 151099018-1099018. https://doi.org/10.3389/FNAGI.2023.1099018

[11] Mulak Agata, Bonaz Bruno. (2015). Brain-gut-microbiota axis in Parkinson's disease.[J]. World journal of gastroenterology, 21(37), 10609-20. https://doi.org/10.3748/wjg.v21.i37.10609

[12] Houser Madelyn C; Tansey Malú G. (2017). The gut-brain axis: is intestinal inflammation a silent driver of Parkinson's disease pathogenesis?. NPJ Parkinson's disease, 3(12), 3. https://doi.org/10.1038/s41531-016-0002-0

[31] Yang X, Dong B, An L, Zhang Q, Chen Y, Wang H. (2021). Ginsenoside Rb1 ameliorates Glycemic Disorder in Mice with High Fat Diet-Induced Obesity via Regulating Gut Microbiota and Amino Acid Metabolism. Frontiers in Pharmacology, 12756491-756491. https://doi.org/10.3389/FPHAR.2021.756491

[32] Bai X, Fu R, Duan Z, Wang P, Zhu Ch. (2021). Ginsenoside Rk3 alleviates gut microbiota dysbiosis and colonic inflammation in antibiotic-treated mice. Food Research International, 146110465-110465. https://doi.org/10.1016/J.FOODRES. 2021.110465

Comment 2:

What are the specific doses of WEG in the low, middle, and high groups in the article, and what is the basis for these doses? The article mentions oral administration of WEG from day 28 to 56. How was this application period determined? In pharmacological experiments, the establishment of dosage and timing requires sufficient justification.

Authors’ responses:

According to the records in Pharmacopoeia of the People's Republic of China 2020, the daily oral dose of ginseng for human should be 3–9 g, and the corresponding administration groups of mice were categorized into low (0.013g/kg body weight per day), middle (0.026g/kg body weight per day), and high groups (0.039g/kg body weight per day). The MPTP-induced chronic injury model is more consistent with Parkinson's disease, and it takes frequent, high-dose injections of MPTP to induce a large amount of DA deficiency in mice, but this deficiency is usually reversed within 3-6 months.

We rationalized the modeling time to 28 days based on the pre-experiment after consulting the literature, and since DA deficits may be reversed at 3-6 months, we aligned the drug administration time with the modeling time to ensure that the experimental improvement originated from the drug and not from automatic recovery of DA in the organism.

Comment 3:

In Figure 6, the IHC staining of WEG is shown at x400 magnification, which is quite large. Are there images available at x100 or x50 magnification? This would provide clearer comparisons.

Authors’ responses:

We have changed Figures 5 and 6 to x200x images to ensure that the images were taken from the same location and can be compared more clearly.

Comment 4:

Please consider streamlining the article, particularly the introduction and discussion sections. Any redundant content can be removed.

Authors’ responses:

We have revised the introductory section and marked the changes in yellow.

Reviewer 3

The study focused on the neuroprotective effects of water extract of ginseng (WEG) on a mouse model of Parkinson's Disease (PD) induced by MPTP (1-methyl-4-phenyl-1,2,3,6-tetrahydropyridine). The authors presented significant results indicating that WEG could ameliorate MPTP-induced PD-like symptoms, improve intestinal damage, reduce α-Syn accumulation, increase TH level, and rectify gut microbiota dysbiosis. The study provides valuable insights into the potential of WEG as a pharmacological treatment for PD. However, several points could be further clarified or improved.

Comment 1:

The authors mentioned that WEG extraction was performed under mild conditions of low temperature and water extraction. It would be interesting to compare these results with those obtained under different conditions. More discussion on the extraction technique might enhance the understanding of the effects of extraction conditions on the efficacy of WEG.

Authors’ responses:

We appreciate the reviewers' comment, but the discussion of our manuscript focuses on the pathological study of WEG to improve PD mice. For different extraction conditions is not the main statement of this manuscript.

Comment 2:

Although the authors suggest that WEG plays a role in resisting PD progression, the exact mechanisms by which WEG exerts its effects are not completely elucidated. Further exploration of this could provide a more detailed understanding of the mechanisms underlying the effects of WEG.

Authors’ responses:

We thank the reviewers for their comments on the mechanism by which WEG exerts its effect, which is what we are most interested in in the research work of this paper. We have several speculations about this, which are also likely to be the focus of our research in the next work plan.

1) WEG ameliorates the pathogenesis of PD by repressing certain mitochondrial apoptosis-related genes.

2) Anti-oxidant effect: WEG may inhibit the levels of certain pro-oxidant factors, thus acting as a neuroprotective agent in the brain.

3) Gene-level effects: PD occurs as a result of multi-gene, multi-step, mutations, and mutations in different genes with different intensities form different tumors. The addition of WEG may have altered the structure of DNA, the genetic material of mice brain nerve cells, reversing the genetic characteristics of brain nerve cells and thus exerting an inhibitory effect on parkinsonism.

Comment 3:

It was mentioned that the high dose of WEG improved PD and also affected the gut microbiota of the mice. More information or discussion on the correlation between dosage and effects would be beneficial.

Authors’ responses:

We discussed this and revised the manuscript.

Comment 4:

For figure 5 and figure 6, a low magnification overview is needed for the structure of the brain to ensure the zoom in image was taken from the same location.

Authors’ responses:

We have changed Figures 5 and 6 to x200x images to ensure that the images were taken from the same location and can be compared more clearly.

Comment 5:

For figure 7B, what's the top band of GAPDH? GAPDH should only have one band.

Authors’ responses:

The GAPDH primary antibody has been reselected in Figure 7B. We have modified the figure 7B.

---

## [Decision Letter · Decision Letter 1]

25 Oct 2023

PONE-D-23-02610R1Water extract of ginseng alleviates parkinsonism in MPTP–induced Parkinson's disease micePLOS ONE

Dear Dr. Zhao,

Thank you for submitting your manuscript to PLOS ONE. After careful consideration, we feel that it has merit but does not fully meet PLOS ONE’s publication criteria as it currently stands. Therefore, we invite you to submit a revised version of the manuscript that addresses the points raised during the review process.

We look forward to receiving your revised manuscript.

Kind regards,

Chun-Hua Wang

Academic Editor

PLOS ONE

Journal Requirements:

Reviewers' comments:

Reviewer's Responses to Questions

**Comments to the Author**

1. If the authors have adequately addressed your comments raised in a previous round of review and you feel that this manuscript is now acceptable for publication, you may indicate that here to bypass the “Comments to the Author” section, enter your conflict of interest statement in the “Confidential to Editor” section, and submit your "Accept" recommendation.

Reviewer #4: All comments have been addressed

2. Is the manuscript technically sound, and do the data support the conclusions?

Reviewer #4: Yes

3. Has the statistical analysis been performed appropriately and rigorously? 

Reviewer #4: No

4. Have the authors made all data underlying the findings in their manuscript fully available?

Reviewer #4: Yes

5. Is the manuscript presented in an intelligible fashion and written in standard English?

Reviewer #4: Yes

6. Review Comments to the Author

Reviewer #4: Dear Author,

please refer to the manuscript. I have provided several comments that required your attention for the necessary changes.

7. PLOS authors have the option to publish the peer review history of their article (what does this mean?). If published, this will include your full peer review and any attached files.

Reviewer #4: No

---

## [Editor Report · Decision Letter 2]

14 Dec 2023

Water extract of ginseng alleviates parkinsonism in MPTP–induced Parkinson's disease mice

PONE-D-23-02610R2

Dear Dr. Zhao,

We’re pleased to inform you that your manuscript has been judged scientifically suitable for publication and will be formally accepted for publication once it meets all outstanding technical requirements.

Kind regards,

Chun-Hua Wang

Academic Editor

PLOS ONE
---

## [Editor Report · Acceptance letter]

11 May 2024

PONE-D-23-02610R2 

PLOS ONE

Dear Dr. Zhao, 

I'm pleased to inform you that your manuscript has been deemed suitable for publication in PLOS ONE. Congratulations! Your manuscript is now being handed over to our production team.

Kind regards, 

on behalf of

Dr. Chun-Hua Wang 

Academic Editor

PLOS ONE